# Variability of BVOC Emissions from Commercially Used Willow (*Salix* spp.) Varieties

**Tomas Karlsson** [1,*]**, Riikka Rinnan** [2]  **and Thomas Holst** [1,2]

[1] Department of Physical Geography and Ecosystem Science, Lund University, Sölvegatan 12, SE-223 62 Lund, Sweden; thomas.holst@nateko.lu.se

[2] Terrestrial Ecology Section, Department of Biology, University of Copenhagen, Universitetsparken 15, DK-2100 Copenhagen Ø, Denmark; riikkar@bio.ku.dk

[*] Correspondence: tomas.karlsson@nateko.lu.se

**Abstract:** Willow (*Salix* spp.) trees are commonly used in short rotation coppices (SRC) to produce renewable energy. However, these plants are also known to emit high concentrations of biogenic volatile organic compounds (BVOCs), which have a large influence on air quality. Many different clones of commercially used *Salix* varieties exist today, but only a few studies have focused on BVOC emissions from these newer varieties. In this study, four varieties commercially propagated for biofuel production have been studied on a leaf-scale in the southern part of Sweden. The trees had either their first or second growing season, and measurements on BVOC emissions were done during the growing season in 2017 from the end of May to the beginning of September. Isoprene was the dominant emitted compound for all varieties but the average emission amongst varieties varied from 4.00 to 12.66 µg $g_{dw}^{-1}$ h$^{-1}$. Average monoterpene (MT) (0.78–1.87 µg $g_{dw}^{-1}$ h$^{-1}$) and sesquiterpene (SQT) emission rates (0.22–0.57 µg $g_{dw}^{-1}$ h$^{-1}$) differed as well among the varieties. Besides isoprene, other compounds like ocimene, linalool and caryophyllene also showed a response to light but not for all varieties. Younger plants had several times higher emissions of non-isoprenoids (other VOCs) than the corresponding 1-year-old trees. The conclusions from this study show that the choice of variety can have a large impact on the regional BVOC emission budget. Genetics, together with stand age, should be taken into account when modelling BVOC emissions on a regional scale, for example, for air quality assessments.

**Keywords:** *Salix*; biofuel plantation; terpenoid emissions; BVOC

## 1. Introduction

The extended use of biofuels is widely promoted to decrease the carbon (C) emissions from fossil fuels, i.e., to fulfil the requirements of the EU directive (2009/28/EC) on renewable energies and to achieve zero net emissions of greenhouse gases in Sweden by the year 2045 [1,2]. In 2017, biofuels alone contributed 25% [3] of Sweden's total energy supply. Most of the energy supply from biofuels are based on 'classical' forest products (pulp industry fuels, wood fuel and sawmill by-products) but logging residues and tree stumps have also been used [4]. However, the contribution to energy production based on agroforestry (i.e., energy crops) is expected to increase to meet the requirements of the EU directive. Besides 'classical' biofuel crops like rapeseed, sugar beets or oil seeds, fast-growing tree species (willow (*Salix* spp.), poplar and hybrid aspen) are increasingly used as energy crops [5], either for direct combustion or for the production of liquid fuels by 'second generation' bioethanol from lignocellulose. Energy crops are currently using 3% of arable land in Sweden [5]. *Salix* trees have been reported to grow on 12,000 ha in Sweden in 2014 [6] but the potential use is estimated at 300,000 ha [5]. The advantages of willow are the high energy content (more than twice that of oat), the ability to clean

up from soils waste water treatment products and cadmium, and the greater increase in the C stock in soil and mulch compared to with annual crops, as willow is grown for 4 years before cutting [7,8].

Willow has been used intensively as energy crop since the 1990s, and varieties have been propagated to increase both biomass production and resistance against weeds and pests [9]. Depending on the climate conditions, different varieties are suitable. For instance, at higher latitudes, such as in the middle and northern parts of Sweden, varieties need to be more resistant to frost, whereas at southern latitudes, trees can suffer from heat damage. There exist no official data on the distribution of the varieties, but studies have shown that Tora has been successfully grown in Sweden, as it gives 40–50% higher yield and has a better resistance to rust compared to older varieties such as L 78183, Orm and Rapp [9]. Other varieties, e.g., Sven and Inger, are suitable to be grown in Sweden, and Inger is also suitable for soils with a low soil water capacity [10]. As *Salix* is easy to propagate, new varieties are continuously propagated by commercial companies that aim at increasing biomass yield and tolerance against insects, plant pests and weeds. Additionally, a change of growing conditions due to climate change might imply that older varieties should be replaced with newer ones, which have been specifically propagated to cope better with drought.

While *Salix* plantations may be a good option for energy crop production, a large-scale land-use change towards *Salix* might have severe impacts on atmospheric chemistry and local air quality. Areas used for short-rotation coppices (SRC) to increase the production of biofuels are most converted from traditional agricultural crops. In contrast to agricultural crops, *Salix* species are regarded as high-emitters of biogenic volatile organic compounds (BVOCs) [11] that are very reactive and can contribute to the production of ozone ($O_3$) and secondary organic aerosols (SOAs) [12–17]. *Salix* species have been shown to emit large amounts of isoprene, with standardized emission rates ranging from 12.5 to 115.0 µg $g_{dw}^{-1}$ $h^{-1}$ [11,18–22]. Monoterpene (MT) emissions from some *Salix* spp. have also been reported [11,18,23], but data for the quantification of compounds other than isoprene and monoterpenes (MTs) are scarce [22].

The large variation of published standardized emission rates for *Salix* spp. indicates an influence of genetic disposition on the BVOC production and emission [24], which has been observed for other species as well [25,26]. Consequently, commercial propagation methods to find better varieties that provide higher biomass yields, increased resistance against plant pests and enhanced competitiveness against weeds might also affect the production and emission of BVOCs.

Here, we analyze leaf-scale BVOC emissions from several varieties of willow that were growing either in field trails or commercially on SRC plantations. We aim to identify the compound spectrum emitted by these varieties, and provide standardized emission rates that can be used in emission inventories and distributed vegetation models to assess the impact of willow plantations on regional air quality.

## 2. Material and Methods

### 2.1. Experimental Sites

Four plots in southern Sweden were used in this study. Two of them (plot 1, 55°52′32.9′′ N 13°1′18.2′′ E and plot 2, 55°52′11.7′′ N 13°1′33.3′′ E, Figure 1A,B) were field trial areas for a commercial company (European Willow Breeding AB) outside Billeberga and stocked with 12–15 different varieties of willow. Two other plots (plot 3, 58°17′09′′ N 12°45′31′′ E and plot 4, 58°16′55′′ N 12°46′20′′ E, Figure 1C,D) were located outside Grästorp ca 300 km north of plots 1 and 2 and used for the commercial production of biomass for energy purposes. The varieties measured on plot 1 were planted in 2014, but were cut down before the growing season in 2016 (Table 1). The land on plot 1 had previously been used for growing other *Salix* varieties before the new establishment of the varieties in 2014 (Table S1). Plot 2 consisted of almost the same varieties as plot 1, but these trees were planted in 2017. Plot 2 had not been used for growing *Salix* before; instead, crops such as cereals, beets and rapes had been growing here until 2016.

**Table 1.** The plot type, size (ha), canopy height (m), varieties, establishment, last harvest and age (months) for the trees.

|  | Plot 1 | Plot 2 | Plot 3 | Plot 4 |
|---|---|---|---|---|
| **Type** | field trial | field trial | biofuel plantation | biofuel plantation |
| **Plot size** | 0.07 ha | 0.07 ha | 5 ha | 9 ha |
| **Canopy height** [1] | 4.5 m | 1.5 m | 2.5 m | 1–1.5 m |
| **Varieties** | Tora, Wilhelm, Ester and Inger | Tora, Wilhelm, Ester and Inger | Tora | Wilhelm |
| **Established** | 2013 | 2017 | 2003 | 2017 |
| **Last harvest** | 2016 | - | 2017 | - |
| **Age of the trees** [1] | 16 months | 5 months | 9 months | 5 months |

[1] Average canopy height and age for the varieties at the last campaign at each plot.

Each variety on plots 1 and 2 was grown on 5 rows that were a few meters long. The rows were separated by ca 0.7 m and the trees had been planted at 0.5 m intervals. The distance between the two plots was approximately 700 m. The mean annual temperature (T) was 7.7 °C (1961–1990 in Svalöv, located 6 km from the plots) and the accumulated annual precipitation was 687 mm (1961–1990 in Svalöv) [27]. The surrounding area was used for growing traditional crops. Only one variety was growing on plot 3 and another one on plot 4. Plot 3 was established in 2003 and harvested before spring 2017. Plot 4 was replanted during spring 2017, since the older variety, established in 1994, was exterminated after harvest in 2016. The trees on plots 3 and 4 were planted in double rows, with 0.75 cm between the rows in the double row. Each double row was separated with 1.25 m and the space between the trees in a row was 0.4 m. The distance between plots 3 and 4 was approximately 1 km. Mean annual T and precipitation for plots 3 and 4 were 6.1 °C (1961–1990 in Gendalen, located 16 km from the plots) and 683 mm (1961–1990 in Grästorp, located 7 km from the plots), respectively [27]. No watering or fertilization were done on any of the plots.

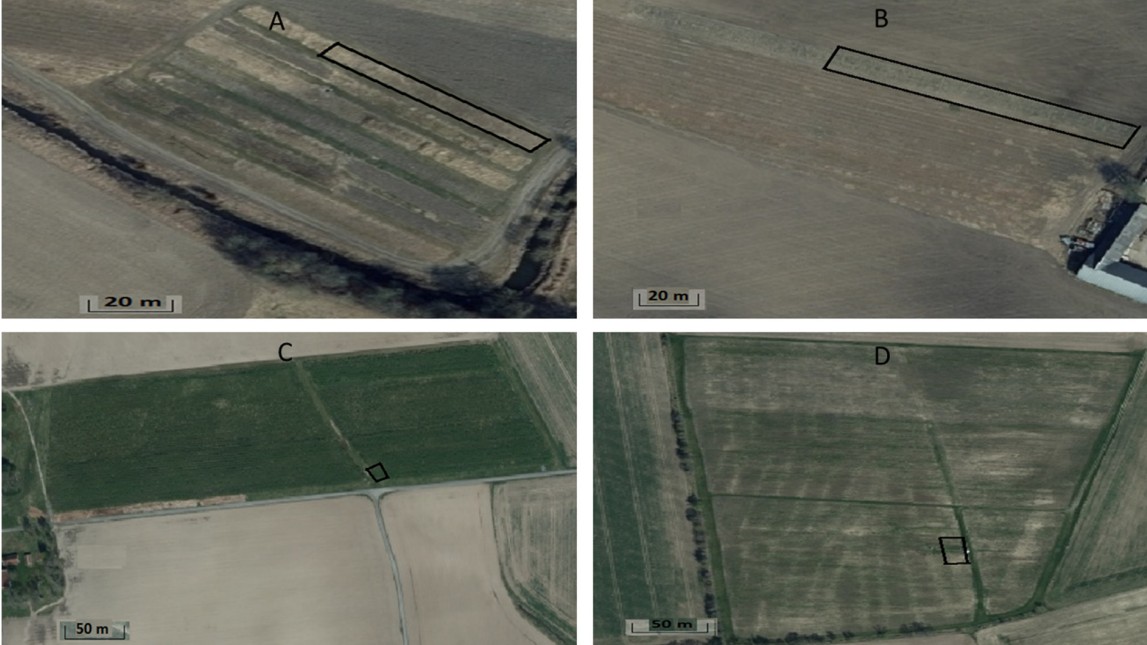

**Figure 1.** The different plantations where the measurements were done. (**A**) Plot 1, the field trial of second year growing varieties established in 2014. (**B**) Plot 2, the field trial of first year growing varieties established in 2017. (**C**) Plot 3, the biofuel plantation of first year growing trees established in 2017. (**D**) Plot 4, the biofuel plantation of first year growing trees established in 2003. The black rectangles indicate the positions of the leaf scale measurement. The photos are modified from Lantmäteriet [28].

### 2.2. Salix Varieties

Near plots 1 and 2, a company (European Willow Breeding AB) has been growing *Salix* trees since 2011. The 4 species chosen are briefly described below and the information was provided by the breeding company. All species have been propagated for commercial use to produce renewable energy. Between 1994 and 2007, regular yield tests were done for new commercial varieties where variety L 78183 was used as a reference with a yield of 100 kg dry weight per plot ($kg_{dw}$ $plot^{-1}$) [9]. This system was discontinued in 2009, and the yield for varieties produced thereafter has been based only on rough estimates.

### S. Tora

Tora is a female hybrid and a cross between the clone L 79069 (*S. schwerinii*) and the variety Orm. The cross was made in 1989 and has shown to be one of the most suitable species for growing in the northern part of Europe, with a yield of 150.5 $kg_{dw}$ $plot^{-1}$ and almost no rust infestation or insect attacks. The estimated growing area in Europe is 5000 ha. Because of the resistance to frost and rust, Tora is one of the most appropriate varieties to grow at northern latitudes, e.g., in Sweden.

### S. Inger

Inger is a female hybrid cross between the clone SW 930887 (*S. triandra* from Siberia) and the variety Jorr. The cross was made in 1994 and gives a high yield in mild or warm climates with a normal water supply. The estimated growing area in Europe is 2000 ha. The yield is 140.5 $kg_{dw}$ $plot^{-1}$.

### S. Wilhelm

Wilhelm is a male hybrid from a cross between the varieties Sherwood and Björn. It was made in 2011 and the biomass productivity from this variety is in between the values for Tora and Inger. The estimated growing area in Europe is 400 ha.

### S. Ester

Ester is a female hybrid and a cross between the variety Linnéa and a clone of "Shrubby willow" (*S. miyabeana*). The cross was made in 2012. The yield from this variety is similar to Inger. Ester is suited for dry and hot climates. Compared to the other species, Ester is almost completely free of leaf beetle attacks but is tasty for game, e.g., roe deer and elk. The estimated growing area in Europe is 200 ha.

### 2.3. BVOC Measurements

All measurements on plots 1 and 2 were executed during 4 campaigns throughout the growing season in 2017. Two species were measured each day, and the length of the campaigns was 4 days each (Table 2). Measurements on plots 3 and 4 were divided into 5 campaigns, and the length varied from 1 to 3 days. In total, 319 sample cartridges were taken, but 20 out of these were lost during GC-MS analyses. Toluene and butylated hydroxytoluene had to be removed from the measurements done by LI-6400XT, and toluene from LI-6400, since huge peaks were seen in the background samples, indicating that they were emitted from the instruments. One of the two unknown sesquiterpene (SQT) compounds could not be completely determined by NIST 8.0 database, and 2 options were suggested, copaene or $\alpha$-cubebene; copaene was chosen.

**Table 2.** All of the campaigns in 2017 and the number of samples taken for each variety. The abbreviations in the parentheses indicate what age the varieties were (e.g., T1 means the first growing season for Tora).

| Plot 1 | 29 & 31 May, 2 & 5 June | 5–6 & 9–10 July | 22, 25–26 & 28 July | 28–31 August | |
|---|---|---|---|---|---|
| Tora (T2) | 7 | 7 | 7 | 6 | |
| Wilhelm (W2) | 7 | 7 | 7 | 6 | |
| Ester (E2) | 7 | 7 | 7 | 6 | |
| Inger (I2) | 7 | 7 | 7 | 7 | |
| **Plot 2** | | | | | |
| Tora (T1) | 7 | 7 | 7 | 7 | |
| Wilhelm (W1) | 7 | 7 | 7 | 6 | |
| Ester (E1) | 7 | 7 | 7 | 5 | |
| Inger (I1) | 7 | 7 | 7 | 3 | |
| **Plot 3** | **15 June** | **15 July** | **1 August** | **7 September** | |
| Tora (T1) | 7 | 14 | 7 | 7 | |
| **Plot 4** | **13 June** | **28 June** | **12–14 July** | **2 August** | **5 September** |
| Wilhelm (W1) | 7 | 14 | 18 | 4 | 7 |

## 2.4. Experimental Setup

Fully expanded sun-exposed leaves were chosen from the upper part of the canopy. Two portable photosynthesis systems (LI-6400/LI-6400XT, LI-COR, Lincoln, NE, USA) with $2 \times 3$ cm$^2$ LED source leaf chambers (6400-02B) were used. The middle part of a *Salix* leaf was inserted into the chamber so that the maximum area of the leaf was used in the chamber. Air was continuously entering the chamber with a flow rate of 500 µmol s$^{-1}$ (approximately 0.7 l min$^{-1}$). This purge air passed through a hydrocarbon trap filter (Alltech, Associates Inc., USA) that contained active carbon and MnO$_2$-coated copper nets to clean the air of BVOCs and O$_3$ before it entered the chamber. The temperature inside the chamber was set to match the expected ambient temperature taken from the weather forecast, and the reference CO$_2$ within the chamber was set to 400 ppm. Relative humidity (RH) inside the chamber was regulated to be close to ambient RH and mostly varied between 40% and 70%. The photosynthesis systems were modified for BVOC measurements on adsorbent cartridges by adding a flow divider at the leaf chamber outlet, which lead one part of the sample air towards the built-in gas analyzer (CO$_2$, H$_2$O) of the photosynthesis system. A second sub-sample of 200 mL min$^{-1}$ was pulled through a sample cartridge (Markes International Limited, Llantrisant, UK) by a battery-operated pump (Pocket Pump, SKC Ltd., Dorset, UK). The sample cartridges were filled in a 2-bed configuration with Tenax TA (porous organic polymer) and Carbograph 1TD (graphitized carbon black) adsorbents. Similar set-ups had been used in other studies before [26,29–31].

Samples were collected at seven light levels (0, 150, 300, 450, 600, 1000 and 1500 µmol m$^2$ s$^{-1}$) in order to generate a light response curve for the BVOC emissions. Measurements started 1 h after enclosing the leaf by the chamber to prevent stress-induced BVOC emissions from affecting the samples [29], and the air from the chamber was sampled for 20 min at 200 mL min$^{-1}$ (total sample volume of 4 L). After switching the light conditions for the next step, 30 min was allowed to pass to allow for the leaf to adapt to the new light conditions before BVOC sampling was continued at the new light level. During the measurements, the net assimilation rates (A, µmol CO$_2$ m$^{-2}$ s$^{-1}$) and transpiration (Tr, mmol H$_2$O m$^{-2}$ s$^{-1}$) were measured. By taking the ratio between A and Tr, water use efficiency (WUE, mmol CO$_2$ mol$^{-1}$ H$_2$O) was calculated. Because of problems with matching the concentrations of CO$_2$ and H$_2$O in the sample cell to those in the reference cell in the leaf chamber, the number of measurements included in the A and WUE calculations had to be reduced to 226. At the end of each light response curve, a background sample was taken from an empty chamber to determine the background concentration of the purge air. Ambient T and RH at canopy level were recorded during

the measurement (CS215, Campbell Scientific, USA). Photosynthetically active radiation (PAR, µmol $m^{-2} s^{-1}$) above the canopy was also measured (Li-190, LI-COR, USA) and together with ambient T and RH, these data were recorded by a logger (CR1000, Campbell Scientific, USA). When sampling was completed, the leaves were harvested and dried for two days at 75 °C to determine the dry weights.

After sampling, the cartridges were sealed with long-term storage caps and stored at 3 °C before being analyzed by TD-GC-MS in the laboratory [32]. Compounds were analyzed in the Enhanced ChemStation (MSD ChemStation E.02.01.1177, Copyright 1989–2010 Agilent Technologies, Inc.) and identified by pure standards (isoprene, 2-methylfuran, toluene, 1-octene, hexanal, furfural, 2-hexanal, p-xylene, o-xylene, α-pinene, camphene, benzaldehyde, β-pinene, myrcene, octanal, cis-3-hexenyl acetate, d-phellandrene, p-cymene, eucalyptol, ocimene, terpinolene, linalool, nonanal, cis-3-hexenyl butaryte, aromadendrene, humulene and nerolidol) or using the NIST 8.0 database. Standard mixtures were prepared in methanol at a concentration of 20 µg $mL^{-1}$ and injected into cartridges under a steady stream of helium (100 mL $min^{-1}$). The standards were run at the beginning and at the end of every batch of samples. The sample concentrations were calculated by using the ratios between the sample peak areas and standard peak areas. To check the linearity of responses for the compounds, calibration curves were generated periodically by serial dilutions of the standard mixture to six concentrations (0–25 µg $mL^{-1}$). The detection limit was based on the background samples. Only sample peaks that had an area twice as large (or more) as the corresponding peaks in the background sample were included in the analysis. To be able to quantify BVOCs for which no standard was available, α-pinene was used for MTs, humulene was used for SQTs and toluene was used for other VOCs (compounds not belonging to terpenoids).

### 2.5. BVOC Emissions and Standardization

The emission rates (*E*) of BVOCs were calculated by using Equation (1), shown below (see e.g., [33])

$$E = (C_2 - C_1) \times F \times m^{-1} \tag{1}$$

where E (µg $g_{dw}^{-1}$ $h^{-1}$) is the emission rate, $C_2$ (µg $l^{-1}$) is BVOC concentration in the samples, $C_1$ (µg $l^{-1}$) is the BVOC concentration in the purge air, F (l $h^{-1}$) is the flow rate of the purge air and m (g) is the dried mass of the leaves. Only compounds that had at least twice as high a concentration in the sample air $C_2$ as in the VOC-filtered purge air, $C_1$, were included in the analysis.

To be able to compare emission rates with other studies, they needed to be standardized, because prevailing environmental factors (T, PAR) govern some of the compounds. This normalization can be done in two ways. For compounds that are light and temperature dependent (e.g., isoprene), Equation (2) was used according to Guenther et al. [34]. The standard values for T and PAR are 303.15 K and 1000 µmol $m^{-2} s^{-1}$.

$$E = E_s \times C_T \times C_L \tag{2}$$

E (µg $g_{dw}^{-1}$ $h^{-1}$) is the actual (measured) emission at the chamber temperature T (K) and PAR (µmol $m^{-2} s^{-1}$). $E_s$ (µg $g_{dw}^{-1}$ $h^{-1}$) is the standardized emission, and $C_T$ and $C_L$ are correction factors for temperature and light as defined by Equations (3) and (4).

$$C_L = \frac{\alpha C_{L1} PAR}{\sqrt{1 + \alpha^2 PAR^2}} \tag{3}$$

where $\alpha$ (=0.0027) and $C_{L1}$ (=1.066) are empirical coefficients [34].

$$C_T = \frac{\exp \frac{C_{T1}(T - T_s)}{R T_s T}}{1 + \exp \frac{C_{T2}(T - T_M)}{R T_s T}} \tag{4}$$

where $C_{T1}$ (=95,000 J mol$^{-1}$), $C_{T2}$ (=230,000 J mol$^{-1}$) and $T_M$ (=314 K) are empirical coefficients, $R$ (=8.314 J K$^{-1}$ mol$^{-1}$) is the universal gas constant and $T_s$ (=303.15 K) is the standard temperature [34]. For the compounds that showed light dependence, a curve was fitted by optimizing the parameters on the right-hand side of Equation (2) to the measured emission values. In this procedure, $C_T$ was kept as a constant and determined by the average T for each variety.

For compounds where emissions are dependent on T alone, Equation (5) can be used.

$$E = E_S \times e^{\beta(T - T_s)} \tag{5}$$

where $E$ (µg g$_{dw}$$^{-1}$ h$^{-1}$) is the actual emission rate at temperature T (K), $E_s$ (µg g$_{dw}$$^{-1}$ h$^{-1}$) is the standard emission rate at the standard temperature $T_s$ (=303.15 K) and $\beta$ (=0.09 K$^{-1}$ for MTs and 0.17 K$^{-1}$ for SQTs) is an empirical constant [34,35].

*2.6. Statistical Analysis*

The significance of the differences between the emissions from the varieties were analyzed by a Kruskal-Wallis test, which compared all varieties within a BVOC group. If this test resulted in a significant *p*-value ($p < 0.05$), then a Mann-Whitney U-test with a Bonferroni correction to account for multiple comparisons was used on each pair of varieties. Differences in the light responses for isoprene, ocimene and caryophyllene among the varieties were analyzed by multiple linear regression.

**3. Results**

*3.1. Climate Data*

The long-term climate data were taken from the Swedish Meteorological and Hydrological Institute (SMHI). The closest weather station for plots 1 and 2 that records T is located in Lund (circa 19 km from plots 1 and 2) and for precipitation is Landskrona (circa 11 km from plots 1 and 2). The T for plots 3 and 4 was recorded in Gendalen and precipitation was measured in Trökörna (circa 9 km from plots 3 and 4). To be able to compare the phenological status between the two sites, growing degree days (GDDs, °C) was calculated as $\Sigma[(T_{max} + T_{min})/2 - T_{base}]$ for each day where $T_{base} = 5$ °C. If the GDD value was <0 °C for a specific day, then it was set as 0 °C. The long-term GDD means between 1987 and 2016 were calculated (Table 3). Comparing the GDDs in 2017 with the long-term means showed that 2017 was similar to previous years, with a maximum GDD difference < 30 °C. Furthermore, all months had higher GDDs at plots 1 and 2 than at plots 3 and 4, especially May to September. This result indicates that the trees at plots 1 and 2 started to grow before the trees at plots 3 and 4. On the other hand, since the campaigns were not done at the same time for the different plots, the phenological status was not necessarily higher for plots 1 and 2. For example, when the first campaign at plots 1 and 2 was done in May, the GDD value was 335.6 °C at this location. Two weeks later in June, when the first campaign was done at plots 3 and 4, the GDD had risen to 365.8 °C for these plots (Table S2).

**Table 3.** The average ambient T (°C), average precipitation (mm) and mean values of growing degree days (GDDs, °C) for each month in 2017. The long-term means for T and GDD were calculated between 1987 and 2016 for all plots. The long-term precipitation for plots 1 and 2 was calculated between 1987 and 2016, but between 1992 and 2016 for plots 3 and 4. The temperature was recorded in Lund, and the precipitation in Landskrona, for plots 1 and 2. For plots 3 and 4, the T from Gendalen and precipitation from Trökörna have been used.

| 2017 plots 1 & 2 | January | February | March | April | May | June | July | August | September |
|---|---|---|---|---|---|---|---|---|---|
| T (°C) | 0.6 | 1.5 | 4.8 | 6.8 | 13.1 | 16.2 | 16.8 | 17.4 | 13.8 |
| Precipitation (mm) | 13.7 | 39.6 | 36.8 | 31.9 | 14.5 | 77.0 | 58.6 | 57.3 | 61.4 |
| GDDs [1] (°C) | 1.3 | 6.8 | 36.1 | 112.2 | 370.9 | 715.2 | 1082.7 | 1478.6 | 1754.5 |
| **Long-term plots 1 & 2** | | | | | | | | | |
| T (°C) | 0.7 | 1.1 | 3.0 | 7.6 | 12.3 | 15.5 | 18.1 | 17.6 | 13.8 |
| Precipitation (mm) | 44.9 | 34.5 | 32.7 | 30.3 | 43.2 | 59.2 | 65.7 | 74.8 | 60.3 |
| GDDs [1] (°C) | 7.7 | 10.9 | 31.2 | 130.0 | 365.3 | 686.9 | 1101.0 | 1502.9 | 1779.8 |
| **2017 plots 3 & 4** | | | | | | | | | |
| T (°C) | −0.3 | 0.1 | 3.1 | 5.3 | 11.7 | 14.9 | 15.9 | 15.1 | 12.0 |
| Precipitation (mm) | 30.6 | 38.7 | 61.5 | 36.7 | 39.5 | 78 | 36.2 | 92.4 | 68.7 |
| GDDs [1] (°C) | 0.5 | 0.8 | 13.6 | 55.1 | 258.6 | 560.8 | 896.4 | 1215.8 | 1433.4 |
| **Long-term plots 3 & 4** | | | | | | | | | |
| T (°C) | −1.3 | −1.0 | 1.4 | 6.2 | 11.1 | 14.4 | 16.8 | 15.7 | 11.8 |
| Precipitation (mm) | 59.4 | 47.7 | 37.8 | 49.7 | 55.3 | 83.4 | 83.5 | 83.9 | 69.9 |
| GDD [1] (°C) | 1.9 | 4.3 | 13.9 | 73.2 | 255.6 | 532.4 | 893.8 | 1229.0 | 1436.8 |

[1] GDD was calculated as $\Sigma[(T_{max} + T_{min})/2 - T_{base}]$ for each day between 1987 and 2016. $T_{base} = 5$ °C, and if $(T_{max} + T_{min})/2 < 0$, then $(T_{max} + T_{min})/2 - T_{base} = 0$.

More precipitation fell throughout the growing season on plots 3 and 4 compared to on plots 1 and 2, and the only months between January and September that had less rain at plots 3 and 4 were February and July (Table 3). The temperature and precipitation for each month in 2017 differed somewhat compared to the long-term means at the sites, which were calculated between 1987 and 2016 (T plots 1–4 and precipitation plots 1 and 2) and between 1992 and 2016 (precipitation plots 3 and 4). For instance, February, March, May and June were circa 0.4–1.8 °C warmer in 2017 than the long-term means at plots 1 and 2 (Table 3 and Figure S4A). February, March and June in 2017 received circa 13–30% more precipitation than the corresponding months between 1987 and 2016 at plots 1 and 2, whilst May got 66% less rain. For plots 3 and 4, April, July and August were colder (circa 0.6–0.9 °C) and January–March, May, June and September were warmer (circa 0.2–1.7 °C) than the long-term means (Table 3 and Figure S4B). March received more rain (>60%) and January, February, April, May and July had less precipitation (circa 19–57%) compared to the means calculated between 1992 and 2016. The lower T and the drier conditions in April 2017 might have slowed down the growing process for the trees on plots 3 and 4 at the beginning of the growing season.

The absolute difference between the average leaf T within the chamber and the average ambient T during the measurements varied from 0.3 to 2.3 °C at plot 1, 0.2 to 2.4 °C at plot 2, 0.7 to 2.3 °C at plot 3 and 0.1 to 2.1 °C at plot 4 (Table S2). The temperature differences between the chamber and ambient air are small and are considered to cause no or little stress to the trees, since the leaf T is close to the prevailing T that the rest of the tree experienced during the measurement.

### 3.2. BVOC Emission

From all campaigns, in total, 193 different peaks could be detected during GC-MS analysis, but only 87 compounds could be identified. The unidentified peaks were named as unknown. The average measured BVOC emission from all varieties and plots for the whole season in 2017 was 26.33 (± 1.54) $\mu g\ g_{dw}^{-1}\ h^{-1}$. The average measured BVOC emission varied campaign-wise from 3.17 to 55.34 $\mu g\ g_{dw}^{-1}\ h^{-1}$, where the highest emission rate was seen during the first campaign (29 May–5 June) and the lowest during the last (5–7 September) (Figure 2). The dominant compound from the trees was isoprene, with an average emission rate of 8.08 (± 14.67) $\mu g\ g_{dw}^{-1}\ h^{-1}$ (Table 4). Isoprene

contributed almost 30% of the total BVOC emissions. The average measured isoprene emission rate for each campaign varied between 0 and 21.61 µg $g_{dw}^{-1}$ $h^{-1}$, where no emissions could be seen during the second campaign (13–15 June) from plots 3 and 4. The highest average measured emission rate was observed during the third campaign (28 June). Except for in the third campaign, the fraction of isoprene was higher in the second part of the season, where it varied from 40% to 70% of the total BVOC emissions.

Fourteen different MTs were emitted, whereof 13 were identified (α-pinene, α-thujene, β-pinene, camphene, d-phellandrene, eucalyptol, γ-terpinene, limonene, linalool, ocimene, p-cymene, terpinolene and 3-carene). Ocimene was the dominant MT and the average rate was 0.54 (± 1.21) µg $g_{dw}^{-1}$ $h^{-1}$. The average MT emission rate for each campaign ranged from 0.10 to 2.59 µg $g_{dw}^{-1}$ $h^{-1}$. The highest rate was emitted during the first campaign (28 May–5 June) and lowest during the last (5–7 September) (Figure 2). The MT fraction reached up to 11% of the total average BVOC emissions at the end of August but was 8% or less during the rest of the campaigns.

Six SQTs were observed, whereof five could be identified (α-farnesene, caryophyllene, copaene, humulene and nerolidol). Nerolidol had the highest emission rate (0.26 ± 0.61 µg $g_{dw}^{-1}$ $h^{-1}$) among all SQTs, which was more than three times higher than that of the second most emitted SQT, which was humulene (0.08 ± 0.28 µg $g_{dw}^{-1}$ $h^{-1}$). The average rates from the group of SQTs were lower than for MTs and varied between 0.007 and 0.74 µg $g_{dw}^{-1}$ $h^{-1}$, where the peaks and the lowest values occurred at same campaigns as for the group of MTs. The contribution from the SQTs to the total BVOC emissions varied from 0.1% to 3.4%. The highest SQT fraction was seen during the seventh campaign (1–2 August) and lowest during the third (28 June).

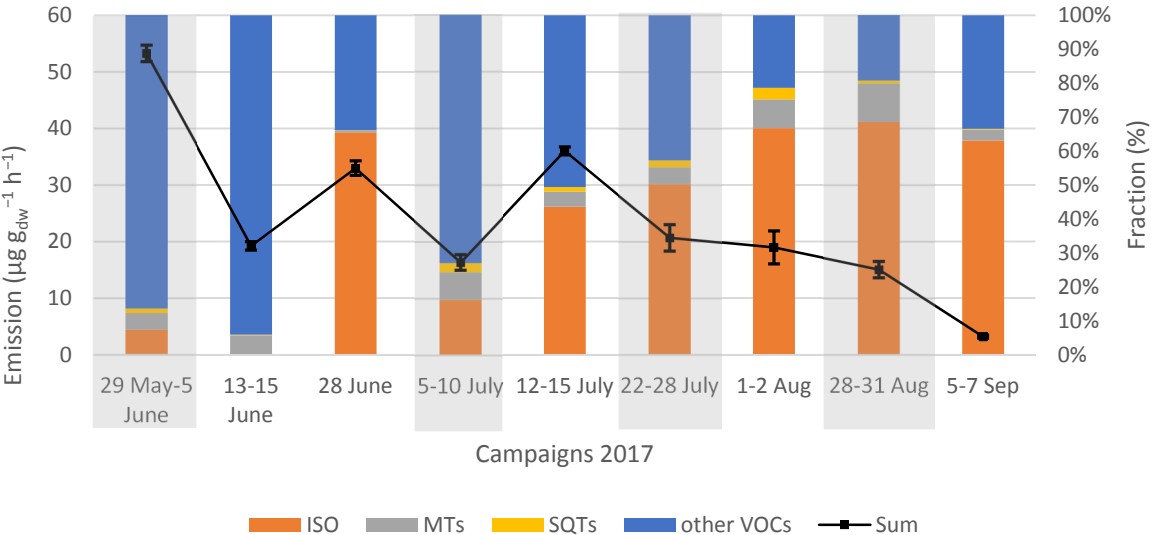

**Figure 2.** Total biogenic volatile organic compound (BVOC) emissions (black square, mean ± standard deviation, µg $g_{dw}^{-1}$ $h^{-1}$, n = 11–56), and the fractions of isoprene (ISO), monoterpenes (MTs), sesquiterpenes (SQTs) and other volatile organic compounds (other VOCs) throughout the season for each campaign. Campaigns performed on plots 1 and 2 (grey boxes) were done from 29 May to 5 June, 5 to 10 July, 22 to 28 July and 28 to 31 August. Campaigns performed on plots 3 and/or 4 were done from 13 to 15 June, on 28 June, from 12 to 15 July, from 1 to 2 August and from 5 to 7 September.

**Table 4.** The measured (M, µg $g_{dw}^{-1}$ $h^{-1}$) and standardized emissions (STD, µg $g_{dw}^{-1}$ $h^{-1}$, n = 299) of the BVOC groups and some of the most abundant identified compounds in each group. The numbers in the parentheses are standard deviations (SD, µg $g_{dw}^{-1}$ $h^{-1}$). No standardization has been done on other VOCs (–).

| BVOC | M ± SD (µg $g_{dw}^{-1}$ $h^{-1}$) | STD ± SD (µg $g_{dw}^{-1}$ $h^{-1}$) |
|---|---|---|
| isoprene | 8.08 (14.67) | 33.21 (53.43) |
| **MTs** | 1.30 (0.68) | 4.40 (2.05) |
| ocimene | 0.54 (1.21) | 2.15 (4.50) |
| limonene | 0.30 (1.74) | 0.84 (4.52) |
| linalool | 0.10 (0.22) | 0.46 (0.86) |
| camphene | 0.07 (1.19) | 0.20 (3.22) |
| β-pinene | 0.07 (0.24) | 0.20 (0.74) |
| **SQTs** | 0.40 (0.29) | 2.51 (2.03) |
| nerolidol | 0.26 (0.61) | 1.67 (4.22) |
| humulene | 0.08 (0.28) | 0.48 (2.08) |
| α-farnesene | 0.02 (0.08) | 0.10 (0.43) |
| **other VOCs** | 16.55 (1.01) | - |
| hexanal | 1.86 (9.61) | - |
| octanal | 1.08 (5.11) | - |
| acetophenone | 0.77 (2.51) | - |
| benzaldehyde | 0.69 (1.87) | - |
| furfural | 0.46 (1.65) | - |
| nonanal | 0.40 (1.38) | |
| 1-hexanol, 2-ethyl- | 0.40 (0.83) | - |
| phenol | 0.37 (0.67) | - |
| p-xylene | 0.30 (0.82) | - |
| decanal | 0.30 (0.65) | - |

The average emissions from other VOCs during each campaign ranged from 1.06 to 45.99 µg $g_{dw}^{-1}$ $h^{-1}$, and the average emissions of all other VOCs were 16.55 ± 1.01 µg $g_{dw}^{-1}$ $h^{-1}$. The highest average rate (45.99 ± 1.36 µg $g_{dw}^{-1}$ $h^{-1}$) was seen during the first campaign and was more than twice as high as the second highest (18.25 ± 2.21 µg $g_{dw}^{-1}$ $h^{-1}$) in the middle of July. The lowest other VOC emission rate was observed during the last campaign. The other VOC fraction varied between 19% and 94%, where the two highest values were observed during the first (86%) and the second (94%) campaigns, while the lowest fraction occurred during the eighth campaign at the end of August (Figure 2). Among the other VOCs, hexanal was the most emitted compound (1.86 ± 9.61 µg $g_{dw}^{-1}$ $h^{-1}$), followed by octanal (1.08 ± 5.11 µg $g_{dw}^{-1}$ $h^{-1}$) and acetophenone (0.77 ± 2.51 µg $g_{dw}^{-1}$ $h^{-1}$) (Table 4). The contribution from these compounds varied between the campaigns. Hexanal contributed almost 53% of the total other VOC emissions in the middle of the season (12–15 July) but <17% during the rest of the campaigns. The emissions of octanal and acetophenone were higher in the beginning of the season, where they contributed circa 3–11% and 4–29% to the total average other VOC emissions, respectively.

3.2.1. Terpenoid Emission Differences between the Varieties

The highest total terpenoid emission rate was from Wilhelm (13.68 ± 5.84 µg $g_{dw}^{-1}$ $h^{-1}$) followed by those from Inger (10.02 ± 3.32 µg $g_{dw}^{-1}$ $h^{-1}$), Ester (7.93 ± 2.73 µg $g_{dw}^{-1}$ $h^{-1}$) and Tora (5.96 ± 2.06 µg $g_{dw}^{-1}$ $h^{-1}$) (Table 5). Both average values of T and PAR were similar for all varieties and varied between 18.9 and 19.0 °C, and 551 and 573 µmol $m^{-2}$ $s^{-1}$, respectively. Isoprene emission was highest for Wilhelm (12.66 ± 20.63 µg $g_{dw}^{-1}$ $h^{-1}$) with a corresponding standardized emission (STD) rate of 50.33 (± 72.63) µg $g_{dw}^{-1}$ $h^{-1}$, but there were no significant differences between the varieties and the measured isoprene emissions (Table 6). However, STD isoprene emission was significantly higher for

Wilhelm compared to for Tora. Isoprene emission exceeded the emissions of both MTs and SQTs for all varieties. Inger emitted the highest amount of MTs, but with high variance. Tora had significantly higher MT emissions than Ester and Wilhelm (Table 6). The average MT emission rate among the varieties varied between 0.80 and 1.87 µg $g_{dw}^{-1}$ $h^{-1}$, which corresponds to the average STD range 3.09–6.00 µg $g_{dw}^{-1}$ $h^{-1}$. Sesquiterpene emissions were significantly higher for Ester, which had twice as high emissions as Wilhelm (0.57 ± 0.44 µg $g_{dw}^{-1}$ $h^{-1}$ vs. 0.22 ± 0.24 µg $g_{dw}^{-1}$ $h^{-1}$). Ester and Inger had a similar average SQT emission rate.

**Table 5.** Upper part: Isoprene, monoterpene (MT), sesquiterpene (SQT) and total terpenoid emissions (µg $g_{dw}^{-1}$ $h^{-1}$, mean ± standard deviation) for the different *Salix* varieties. Middle part: Standardized (STD) emission rates (µg $g_{dw}^{-1}$ $h^{-1}$, mean ± standard deviation). Lower part: Average T (°C), photosynthetically active radiation (PAR, µmol $m^{-2}$ $s^{-1}$) and relative humidity (RH) (%) in the measurement cuvette.

| | Tora, n = 90 | Wilhelm, n = 104 | Ester, n = 53 | Inger, n = 52 |
|---|---|---|---|---|
| isoprene (µg $g_{dw}^{-1}$ $h^{-1}$) | 4.00 (7.05) | 12.66 (20.63) | 6.11 (9.06) | 7.77 (11.65) |
| MTs (µg $g_{dw}^{-1}$ $h^{-1}$) | 1.56 (0.62) | 0.80 (0.28) | 1.25 (1.01) | 1.87 (1.24) |
| SQTs (µg $g_{dw}^{-1}$ $h^{-1}$) | 0.40 (0.28) | 0.22 (0.24) | 0.57 (0.44) | 0.56 (0.26) |
| Sum (µg $g_{dw}^{-1}$ $h^{-1}$) | 5.96 (2.06) | 13.68 (5.84) | 7.93 (2.73) | 10.02 (3.32) |
| STD isoprene (µg $g_{dw}^{-1}$ $h^{-1}$) | 17.99 (27.18) | 50.53 (72.63) | 25.84 (34.36) | 32.75 (47.82) |
| STD MTs (µg $g_{dw}^{-1}$ $h^{-1}$) | 5.84 (1.90) | 3.09 (0.99) | 3.44 (2.35) | 6.00 (3.21) |
| STD SQTs (µg $g_{dw}^{-1}$ $h^{-1}$) | 2.43 (1.63) | 1.30 (1.53) | 3.76 (3.20) | 3.71 (1.91) |
| T (°C) | 19.0 (2.1) | 19.0 (2.0) | 18.9 (2.2) | 19.0 (2.3) |
| PAR (µmol $m^{-2}$ $s^{-1}$) | 571 (487) | 551 (477) | 561 (477) | 573 (494) |
| RH (%) | 61.5 (13.0) | 53.9 (16.9) | 48.2 (9.5) | 51.4 (8.9) |

**Table 6.** The *p*-values from statistical tests for differences between varieties in measured (upper) and STD (lower) isoprene, total MT and SQT emissions. The *p*-values originate from pairwise Mann-Whitney U-tests. The significance level after applying a Bonferroni correction is circa 0.008 (0.05/6 ≈ 0.008). All *p*-values written in bold indicate significant differences.

| | isoprene | | | | STD isoprene | | |
|---|---|---|---|---|---|---|---|
| **Variety** | Wilhelm | Ester | Inger | **Variety** | Wilhelm | Ester | Inger |
| Tora | 0.015 | 0.768 | 0.330 | Tora | **0.007** | 0.802 | 0.343 |
| Wilhelm | | 0.033 | 0.299 | Wilhelm | | 0.024 | 0.217 |
| Ester | | | 0.218 | Ester | | | 0.254 |
| | **MTs** | | | | **STD MTs** | | |
| **Variety** | Wilhelm | Ester | Inger | **Variety** | Wilhelm | Ester | Inger |
| Tora | **0.006** | **<0.001** | 0.108 | Tora | **0.005** | **<0.001** | 0.064 |
| Wilhelm | | 0.107 | 0.632 | Wilhelm | | 0.030 | 0.685 |
| Ester | | | 0.045 | Ester | | | 0.026 |
| | **SQTs** | | | | **STD SQTs** | | |
| **Variety** | Wilhelm | Ester | Inger | **Variety** | Wilhelm | Ester | Inger |
| Tora | **0.006** | 0.892 | **<0.001** | Tora | **0.003** | 0.652 | **<0.001** |
| Wilhelm | | 0.051 | **<0.001** | Wilhelm | | 0.070 | **<0.001** |
| Ester | | | **0.001** | Ester | | | **<0.001** |

Ocimene was the dominant MT emitted among all *Salix* varieties (0.03–1.08 µg $g_{dw}^{-1}$ $h^{-1}$) and made up almost 70% and 50% of the total MT emissions from Tora and Wilhelm, respectively (Figure 3). Ocimene was emitted less by Ester, from which the contribution was lower than 3%. Instead, camphene (31%) contributed most of the MTs from Ester, followed by limonene (29%). On the other hand, the emissions of camphene were negligible for the other varieties. Limonene was the second most

abundant MT (0.14–0.78 µg $g_{dw}^{-1}$ $h^{-1}$) and contributed 11–41% of the total MT rate among the varieties. The lowest MT emission rate was seen from Wilhelm, and the highest from Inger. Inger was the only variety that emitted α-thujene, and Tora was the only variety that did not emit terpinolene.

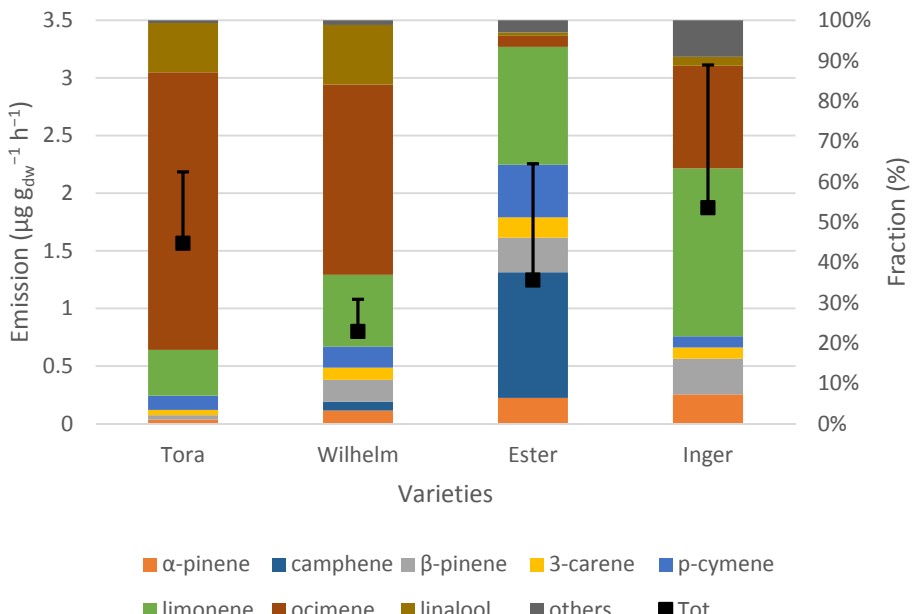

**Figure 3.** Total MT emissions (black square, µg $g_{dw}^{-1}$ $h^{-1}$, mean + standard deviation, n = 52–104) and the contribution from each MT for the different varieties. Others includes d-phellandrene, terpinolene, γ-terpinene and one unknown compound.

Nerolidol was the most prominent SQT and the average emission rate varied from 0.12 to 0.47 µg $g_{dw}^{-1}$ $h^{-1}$ (Figure 4). The contribution from nerolidol was higher than 45% for all varieties and exceeded the contribution from each of the other SQTs. Humulene was the second most emitted SQT (0.04–0.17 µg $g_{dw}^{-1}$ $h^{-1}$) and contributed 7–30% of the total SQT rate. All SQTs observed in this study were seen in emissions from all varieties, except for one unknown, which was not emitted by Tora, and caryophyllene, which was not emitted by Wilhelm.

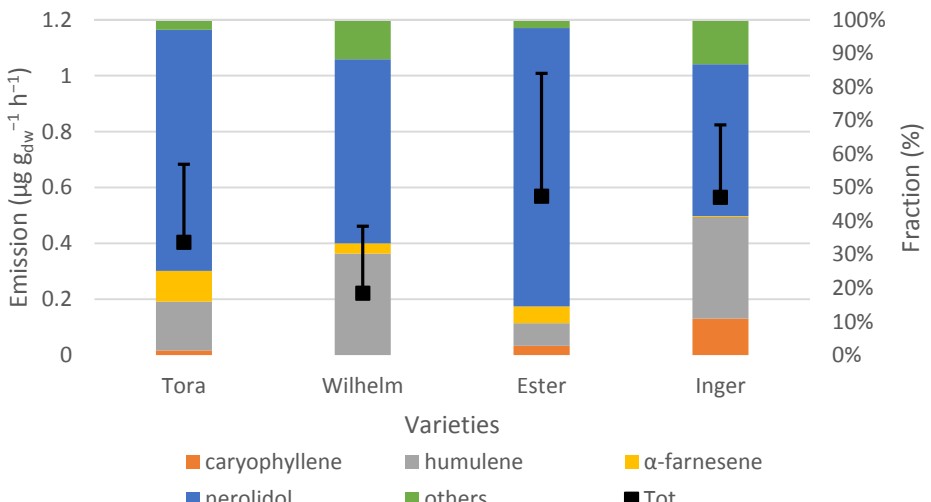

**Figure 4.** Total SQT emissions (black square, µg $g_{dw}^{-1}$ $h^{-1}$, mean + standard deviation, n = 52–104) and the contribution from each SQT for the different varieties. Others includes copaene and one unknown compound.

### 3.2.2. The Responses of Terpenoid Emission, Net Assimilation and Water Use Efficiency to Different Light Levels

The average total terpenoid emission rate increased, in general, with increasing PAR until the highest PAR level of the tested light response curves, 1500 μmol m$^{-2}$ s$^{-1}$, which is explained by the increase in isoprene emission, but this pattern was not completely consistent for Ester and Inger. Terpenoid emissions from Ester had already peaked at a light level of 450 μmol m$^{-2}$ s$^{-1}$ (12.94 ± 3.42 μg g$_{dw}$$^{-1}$ h$^{-1}$), partly due to high emissions of MTs (Figure 5C). Wilhelm emitted higher average terpenoid emissions when PAR varied between 300 and 1500 μmol m$^{-2}$ s$^{-1}$ compared to the other varieties, and the average total terpenoid values for Wilhelm ranged from 0.74 to 34.02 μg g$_{dw}$$^{-1}$ h$^{-1}$, when PAR ranged from 0 to 1500 μmol m$^{-2}$ s$^{-1}$ (Figure 5B). The corresponding emissions from the other three varieties varied between 0.63 and 19.56 μg g$_{dw}$$^{-1}$ h$^{-1}$. Thus, Wilhelm had an average terpenoid emission rate almost twice as high as the second highest (Inger) when PAR peaked. Tora had lower terpenoid emissions (0.68–13.88 μg g$_{dw}$$^{-1}$ h$^{-1}$) than the rest of the varieties at all PAR levels, except at 1500 μmol m$^{-2}$ s$^{-1}$, where Ester had the lowest emission rate (11.34 ± 3.85 μmol m$^{-2}$ s$^{-1}$).

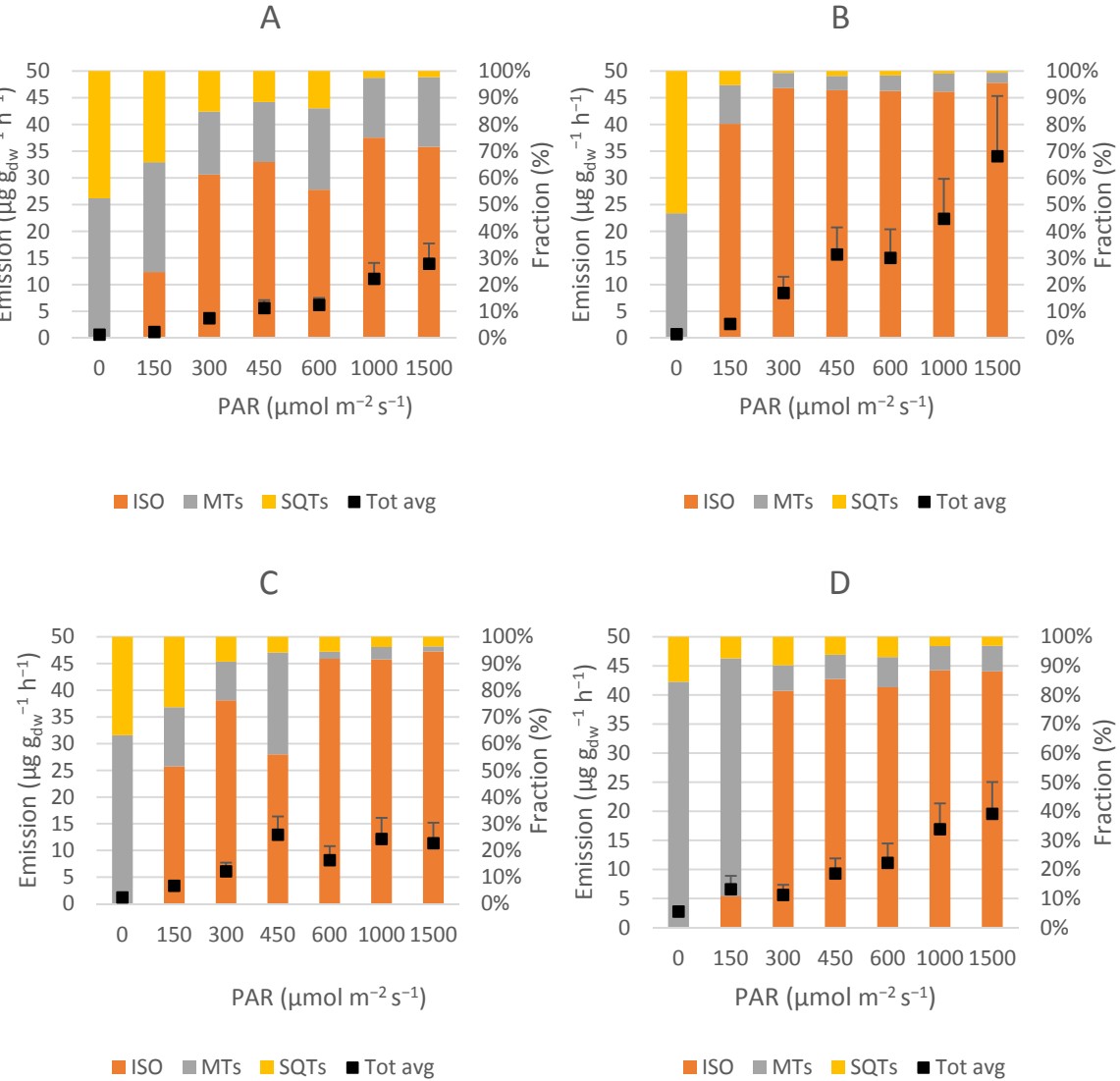

**Figure 5.** Total terpenoid emissions (black squares, μg g$_{dw}$$^{-1}$ h$^{-1}$, mean + standard deviation, n = 7–16) and the fraction of isoprene (ISO), MTs and SQTs for different PAR values (μmol m$^{-2}$ s$^{-1}$). (**A**) Tora, (**B**) Wilhelm, (**C**) Ester and (**D**) Inger.

Isoprene

Isoprene dominated over the other terpenoids and contributed > 50% of the total terpenoid emissions for all varieties when PAR was equal to 300 µmol m$^{-2}$ s$^{-1}$ or more. Isoprene responded faster to the increasing light levels for Wilhelm compared to the other varieties, and the response was significantly higher compared to all the other varieties (Figure 6). Isoprene reached >75% of the total terpenoid emission rate, already at 150 µmol m$^{-2}$ s$^{-1}$, for Wilhelm, and increased to >90% when PAR varied between 300 and 1500 µmol m$^{-2}$ s$^{-1}$ (Figure 5B). The isoprene fraction for Inger did not change between 300 and 1500 µmol m$^{-2}$ s$^{-1}$, and contributed 80–90% of the total terpenoid emissions. Isoprene emission peaked at 1500 µmol m$^{-2}$ s$^{-1}$ for all varieties except Ester, where the average isoprene emission rate reached a maximum at 1000 µmol m$^{-2}$ s$^{-1}$ (Figure 6). At a PAR level of 1500 µmol m$^{-2}$ s$^{-1}$, the highest average isoprene emissions were observed for Wilhelm (32.52 ± 33.81 µg g$_{dw}$$^{-1}$ h$^{-1}$), which was more than three times higher than those for Tora. The fitted curves showed that Ester and Inger responded in the same way up to circa 450 µmol m$^{-2}$ s$^{-1}$. Thereafter, the emission rates from Ester leveled out faster than the others and approached the isoprene emission rates from Tora. Values related to the fitted curves can be seen in Table S3.

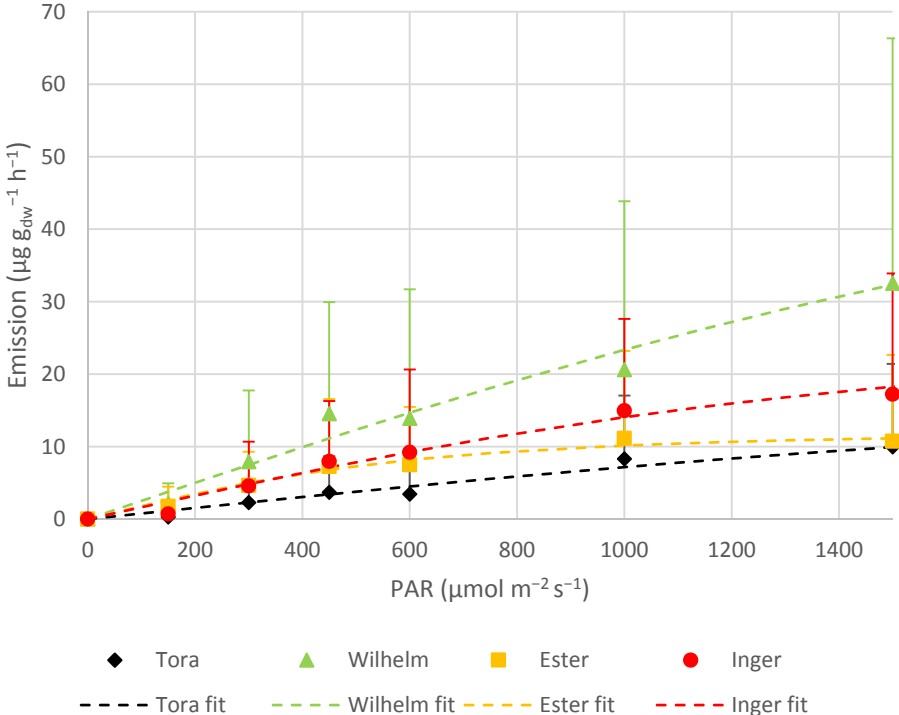

**Figure 6.** Isoprene emission rates (µg g$_{dw}$$^{-1}$ h$^{-1}$, mean + standard deviation, n = 7–16) and fitted curves according to Equation (2) for the *Salix* varieties at different PAR values (µmol m$^{-2}$ s$^{-1}$).

Monoterpenes

Tora was the only variety that constantly had increased MT emissions with increasing PAR (Figure 5A). This trend is mainly due to the light-response for ocimene and linalool, which were the only MTs that showed light dependence. This light dependence was observed for Tora, Wilhelm and Inger, but not for Ester (Figure 7). The average emission of ocimene increased at all PAR steps for Tora (0–2.90 µg g$_{dw}$$^{-1}$ h$^{-1}$) and Inger (0–1.32 µg g$_{dw}$$^{-1}$ h$^{-1}$), whereas it reached 0.75 µg g$_{dw}$$^{-1}$ h$^{-1}$ at 1000 µmol m$^{-2}$ s$^{-1}$ for Wilhelm and leveled out thereafter. The light response of ocimene for Tora was significantly higher compared to for the other varieties. The fitted curves showed that emissions of ocimene from Wilhelm and Inger responded in the same way up to 450 µmol m$^{-2}$ s$^{-1}$, but afterwards, Inger had a steeper response with PAR than Wilhelm. The average emission of linalool from Tora varied from 0 to 0.41 µg g$_{dw}$$^{-1}$ h$^{-1}$, and the light response was significantly higher compared to that for the

others (Figure 8). The trend for Inger was less clear, even if the average emission rates seemed to level out when PAR exceeded 1000 µmol m$^{-2}$ s$^{-1}$. The MT fraction of the terpenoid emissions had a similar pattern for the four varieties. As expected, a substantial contribution to the total terpenoid emissions came from MTs (47–85%) when PAR was equal to 0 µmol m$^{-2}$ s$^{-1}$ for all varieties (Figure 5A–D). When PAR increased to 150 µmol m$^{-2}$ s$^{-1}$, Tora and Inger still had a considerable contribution of MT emissions (41% and 82% respectively), whilst the fractions for Wilhelm and Ester were 14% and 22%, respectively. At the higher light levels, the MT fraction was mainly <10% due to the light-induced isoprene emissions, except for Ester at 450 µmol m$^{-2}$ s$^{-1}$, and for Tora.

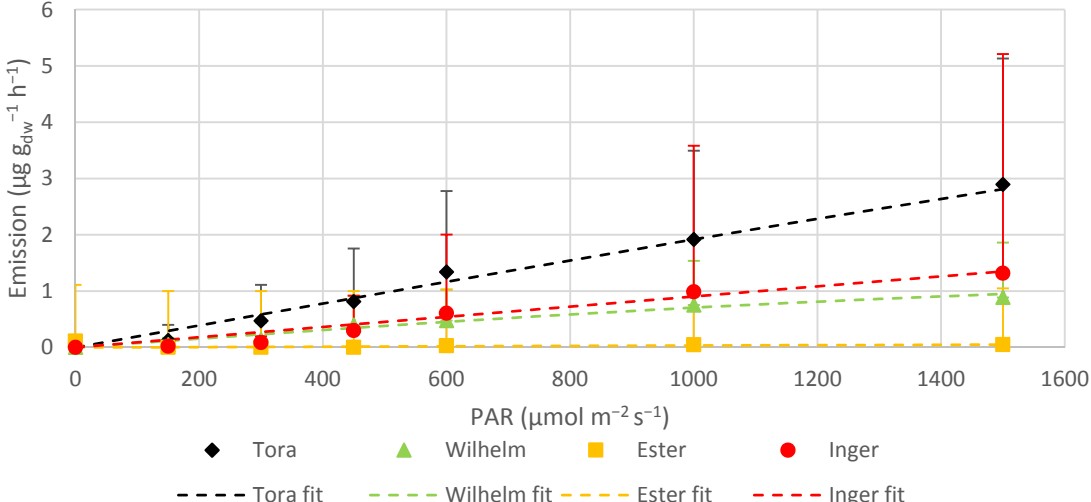

**Figure 7.** The light response of ocimene emissions (µg g$_{dw}$$^{-1}$ h$^{-1}$, mean + standard deviation, n = 7–16) to different PAR values (µmol m$^{-2}$ s$^{-1}$) for the *Salix* varieties.

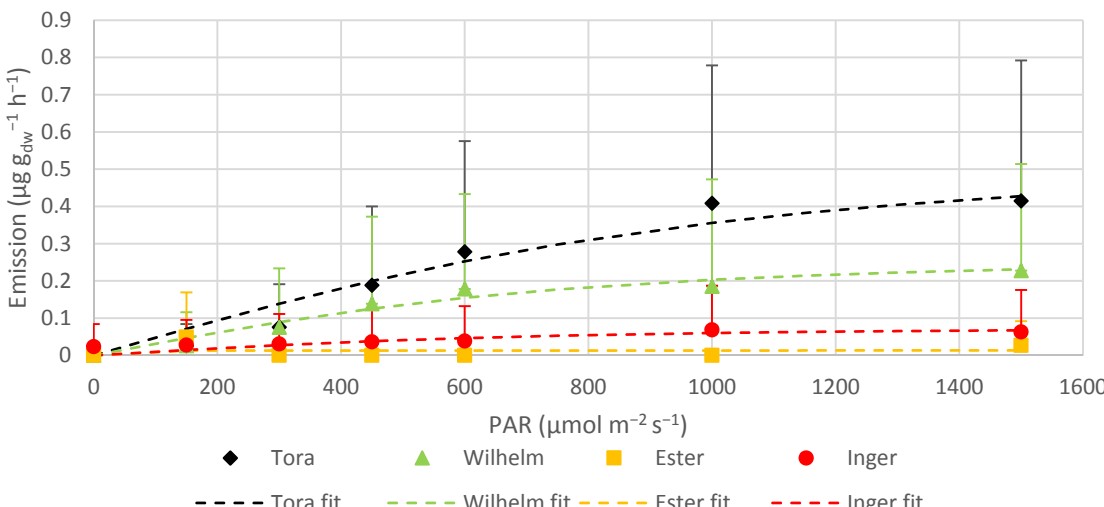

**Figure 8.** The light response of linalool emissions (µg g$_{dw}$$^{-1}$ h$^{-1}$, mean + standard deviation, n = 7–16) to different PAR values (µmol m$^{-2}$ s$^{-1}$) for the *Salix* varieties.

Sesquiterpenes

The emissions of SQTs peaked at both low PAR values (e.g., darkness) and at higher PAR (e.g., 600 µmol m$^{-2}$ s$^{-1}$) among the varieties. Caryophyllene seemed to be influenced by light, but this was only clear for Inger (Figure 9), as its light response for caryophyllene differed significantly to that for the other varieties, and the average emissions increased from 0 to 0.14 µg g$_{dw}$$^{-1}$ h$^{-1}$ when PAR varied between 0 and 1500 µmol m$^{-2}$ s$^{-1}$. The SQT fractions were in general larger for light levels up to 300 µmol m$^{-2}$ s$^{-1}$ (Figure 5A–D). When PAR was equal to 0, SQTs contributed 15–53% of the terpenoid

emission rate. At 150 μmol m$^{-2}$ s$^{-1}$, only Tora and Ester showed a significant contribution of SQTs (34% and 26% respectively) of the total terpenoid emission rate. The contribution from SQTs did not reach above 16% when PAR values were >150 μmol m$^{-2}$ s$^{-1}$. The SQT fraction seemed to decrease with higher PAR values because of increasing isoprene rates. The highest average emissions of SQTs for the different light levels reached up to 0.89 (± 0.49) μg g$_{dw}$$^{-1}$ h$^{-1}$, which was emitted by Ester when PAR was 150 μmol m$^{-2}$ s$^{-1}$. Even the third highest SQT rate (0.76 ± 0.57 μg g$_{dw}$$^{-1}$ h$^{-1}$ at 450 μmol m$^{-2}$ s$^{-1}$) was observed from Ester. Wilhelm emitted lower SQT rates at all light levels, except at zero μmol m$^{-2}$ s$^{-1}$.

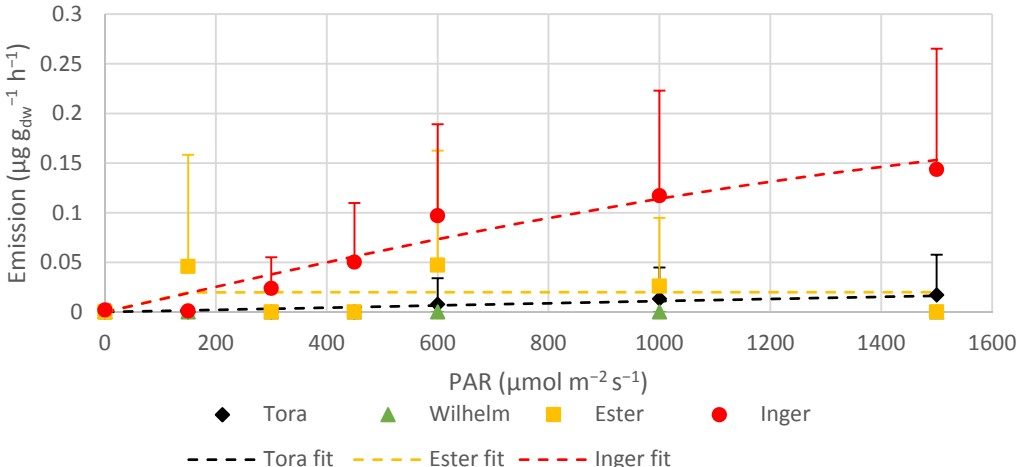

**Figure 9.** The light response of caryophyllene (μg g$_{dw}$$^{-1}$ h$^{-1}$, mean + standard deviation, n = 8) to different PAR values (μmol m$^{-2}$ s$^{-1}$) for the *Salix* varieties.

Net Assimilation (A) and Water Use Efficiency (WUE)

Tora (7.10–17.80 μmol CO$_2$ m$^{-2}$ s$^{-1}$), Wilhelm (7.46–17.15 μmol CO$_2$ m$^{-2}$ s$^{-1}$) and Inger (7.84–17.29 μmol CO$_2$ m$^{-2}$ s$^{-1}$) had a similar A when they were exposed to light (Table 7). The lowest A was observed for Ester, which varied from 5.36 to 12.19 μmol CO$_2$ m$^{-2}$ s$^{-1}$. The WUE was higher for Ester (10.15–15.08 mmol CO$_2$ mol$^{-1}$ H$_2$O) at all light levels compared to for the other varieties (5.55–9.53 mmol CO$_2$ mol$^{-1}$ H$_2$O) and almost twice as high for some PAR values than for the other varieties. For the highest PAR values, Tora was the variety that had lowest WUE. Thus, the water loss due to transpiration was larger for Tora compared to for the others when PAR reached 1000 μmol m$^{-2}$ s$^{-1}$ or more. In contrast, the water loss from Ester was only 35–60% of the loss from the other varieties.

**Table 7.** Upper part: Net assimilation rate (A, μmol CO$_2$ m$^{-2}$ s$^{-1}$, mean ± standard deviation, n = 6–14) at different PAR values (μmol m$^{-2}$ s$^{-1}$) for the varieties. Lower part: Water use efficiency (WUE, mmol CO$_2$ mol$^{-1}$ H$_2$O, mean ± standard deviation, n = 6–14) at different PAR values.

| PAR (μmol m$^{-2}$ s$^{-1}$) | 150 | 300 | 450 | 600 | 1000 | 1500 |
|---|---|---|---|---|---|---|
| Tora (μmol CO$_2$ m$^{-2}$ s$^{-1}$) | 7.10 (1.65) | 10.87 (3.12) | 13.31 (2.70) | 15.00 (3.33) | 16.96 (3.34) | 17.80 (3.58) |
| Wilhelm (μmol CO$_2$ m$^{-2}$ s$^{-1}$) | 7.46 (1.31) | 11.24 (1.87) | 13.75 (2.84) | 14.31 (3.16) | 16.98 (4.48) | 17.15 (5.71) |
| Ester (μmol CO$_2$ m$^{-2}$ s$^{-1}$) | 5.36 (1.02) | 8.65 (1.17) | 9.84 (1.23) | 10.48 (1.66) | 12.12 (2.29) | 12.19 (3.53) |
| Inger (μmol CO$_2$ m$^{-2}$ s$^{-1}$) | 7.84 (2.22) | 11.53 (2.93) | 13.53 (3.69) | 14.49 (3.75) | 16.20 (5.33) | 17.29 (5.42) |
| Tora (mmol CO$_2$ mol$^{-1}$ H$_2$O) | 6.67 (5.42) | 6.57 (3.30) | 6.57 (2.36) | 7.69 (4.28) | 7.12 (2.68) | 6.56 (2.32) |
| Wilhelm (mmol CO$_2$ mol$^{-1}$ H$_2$O) | 6.81 (4.73) | 6.65 (1.67) | 8.38 (4.49) | 7.45 (1.75) | 9.53 (3.89) | 8.00 (2.27) |
| Ester (mmol CO$_2$ mol$^{-1}$ H$_2$O) | 10.15 (8.42) | 12.57 (12.64) | 11.85 (11.81) | 15.08 (11.32) | 11.33 (11.49) | 12.80 (13.58) |
| Inger (mmol CO$_2$ mol$^{-1}$ H$_2$O) | 5.55 (2.51) | 7.47 (2.89) | 8.32 (4.03) | 8.45 (3.38) | 8.32 (3.72) | 8.49 (3.19) |

### 3.2.3. Comparison between Ages

All the younger individuals (T1, W1, E1 and I1) showed higher emission rates of other VOCs compared to the 1-year-old trees (T2, W2, E2 and I2). Especially, the saplings of variety Ester (E0,

46.74 ± 1.18 µg $g_{dw}^{-1}$ $h^{-1}$) emitted 19 times more other VOCs than the older trees of variety Ester (E1, 2.46 ± 0.07 µg $g_{dw}^{-1}$ $h^{-1}$) (Table 8). The high emission rate of other VOCs for E0 is due to a large contribution from hexanal (4.92 ± 6.53 µg $g_{dw}^{-1}$ $h^{-1}$), furfural (3.07 ± 3.93 µg $g_{dw}^{-1}$ $h^{-1}$), benzaldehyde (3.20 ± 4.56 µg $g_{dw}^{-1}$ $h^{-1}$), octanal (3.00 ± 4.79 µg $g_{dw}^{-1}$ $h^{-1}$) and acetophenone (2.46 ± 3.97 µg $g_{dw}^{-1}$ $h^{-1}$) (Table S4). Hexanal contributed substantially for W0 (4.23 ± 17.80 µg $g_{dw}^{-1}$ $h^{-1}$), but less for T0 (1.03 ± 4.88 µg $g_{dw}^{-1}$ $h^{-1}$) and I0 (0.86 ± 1.72 µg $g_{dw}^{-1}$ $h^{-1}$), to the other VOC emissions. The contribution of octanal to the other VOC emissions was high for T0 (3.15 ± 10.16 µg $g_{dw}^{-1}$ $h^{-1}$) but less than 0.50 µg $g_{dw}^{-1}$ $h^{-1}$ for W0 and I0. All the measured and the STD emission rates for MTs and SQTs were higher for the younger than for the older trees, except for Wilhelm (Table 8). Additionally, for MTs and SQTs, Ester was the variety with largest differences between the ages. The emissions for E0 were 10–11 times higher compared to E1. Camphene (0.79 ± 3.96 µg $g_{dw}^{-1}$ $h^{-1}$) and limonene (0.60 ± 1.58 µg $g_{dw}^{-1}$ $h^{-1}$) were the dominant MTs for E0, whereas nerolidol (0.86 ± 1.27 µg $g_{dw}^{-1}$ $h^{-1}$) was the dominant SQT.

**Table 8.** Upper part: Measured emission rates (µg $g_{dw}^{-1}$ $h^{-1}$, mean ± standard deviation, n = 24–77) for isoprene, MTs, SQTs and other VOCs for the different Salix varieties and ages. Lower part: The standardized (STD) emission rate (µg $g_{dw}^{-1}$ $h^{-1}$, mean ± standard deviation, n = 26–77) for isoprene, MTs, and SQTs for the different varieties and ages.

| Variety | Tora | | Wilhelm | | Ester | | Inger | |
|---|---|---|---|---|---|---|---|---|
| Age of trees | 1 | 2 | 1 | 2 | 1 | 2 | 1 | 2 |
| isoprene | 4.21 (7.72) | 3.65 (4.93) | 14.11 (22.13) | 8.53 (14.85) | 5.48 (8.36) | 6.97 (9.74) | 6.61 (9.29) | 8.77 (13.26) |
| MTs | 1.80 (0.69) | 1.02 (0.39) | 0.74 (0.26) | 0.95 (0.34) | 2.33 (1.43) | 0.22 (0.18) | 2.52 (1.69) | 1.32 (0.64) |
| SQTs | 0.52 (0.33) | 0.14 (0.09) | 0.16 (0.18) | 0.40 (0.37) | 1.03 (0.62) | 0.09 (0.08) | 0.79 (0.35) | 0.38 (0.12) |
| other VOCs | 20.91 (1.05) | 3.88 (0.14) | 20.95 (1.50) | 5.87 (0.27) | 46.74 (1.18) | 2.46 (0.07) | 15.70 (0.94) | 4.70 (0.13) |
| STD isoprene | 17.22 (27.74) | 19.76 (25.75) | 52.74 (76.08) | 38.38 (57.38) | 27.47 (37.64) | 24.27 (30.79) | 32.27 (44.84) | 33.16 (50.22) |
| STD MTs | 5.82 (2.05) | 4.50 (1.52) | 2.57 (0.81) | 4.31 (1.38) | 6.36 (3.29) | 0.72 (0.05) | 6.81 (3.96) | 5.30 (0.74) |
| STD SQTs | 3.04 (1.90) | 1.10 (0.62) | 0.78 (0.79) | 2.82 (0.47) | 7.37 (4.49) | 0.28 (0.05) | 5.55 (2.68) | 2.17 (0.36) |

## 4. Discussion

Wilhelm was the variety that emitted the highest rate of terpenoids. Most of this emission (circa 90%) came from isoprene. In fact, Wilhelm emitted over three times more isoprene than Tora and almost twice as much as Ester and Inger. However, when comparing MTs and SQTs, Wilhelm had the lowest emissions. The average emissions of isoprene and SQTs were almost the same for Ester and Inger. The pathways of producing BVOCs have been studied and disentangled to a certain extent. Even if it is not fully understood, studies have shown that there is some linkage between the productions of these compounds [36,37]. The originating substrates responsible for the end products (e.g., isoprene, MTs and SQTs) are shared and divided into the separate pathways, which could be one of the explanations why Wilhelm emits lower amounts of MTs and SQTs, but more isoprene. The average T and PAR values within the chambers were almost the same for the varieties, indicating that the different emission rates among the varieties are related to other differences in the environment, or genetic variation. Genetic diversity was concluded by van Meeningen et al. [30] to be more important than, e.g., local growing conditions, when studying spruce BVOC emissions. Hence, for one specific species, the BVOC emission rates can differ among the varieties or clones. This difference is not always accounted for in models and should not be discarded when improving modelling for upscaling BVOC emissions.

The A was similar for Tora, Wilhelm and Inger (circa 13.5 µmol $CO_2$ $m^{-2}$ $s^{-1}$), reflecting that they are equally good at biomass production in the prevailing conditions in this study. Ester had circa 25% lower A, showing less productivity than the others. Despite the lower A, Ester showed a better ability to utilize water for producing biomass when photosynthesis occurred. The values of WUE related to Ester were up to twice as large compared to the others for some PAR values, which means that Ester lost less than 40% of the water. Therefore, Ester is more suitable for hot and dry climates and it outcompetes the other varieties in regions warmer and drier than southern Sweden. The maximum A for Salix trees has been reported to range from 20 to 30 µmol $CO_2$ $m^{-2}$ $s^{-1}$ [38]. The varieties in this

study had, in general, lower A, but they were able to assimilate more than 20 μmol $CO_2$ $m^{-2}$ $s^{-1}$ when PAR reached 1000 or 1500 μmol $m^{-2}$ $s^{-1}$.

As expected [31,34,39–41], isoprene increased with increasing PAR levels. Studies have also shown that the emission rates of isoprene have a hyperbolic relationship with PAR [34,40,42,43]. Tora, Ester and Inger showed a similar trend, where the emission rates levelled out for the higher light levels. Ester was the only variety that peaked at 1000 μmol $m^{-2}$ $s^{-1}$. Since no obvious damages could be seen on the leaves, this result indicates that the leaves belonging to Ester were saturated at lower light levels and could not utilize and respond to the highest PAR level like the other varieties. On the other hand, isoprene emission from Wilhelm continued to increase and showed no trend towards levelling out. Even though Wilhelm and Tora share similar ancestors from the breeding program, Tora is closer to Ester and Inger when it comes to isoprene emission. The photolysis of BVOCs and $NO_x$ can lead to the production of $O_3$ and PAN [44,45], which are harmful for humans and vegetation at high concentrations [46–49]. Isoprene has been shown to be able to increase $O_3$ and PAN [44,50], which makes Wilhelm less preferable in high-$NO_x$ environments compared to the other varieties. A major part of land cover in Sweden is boreal forest, whereof most is spruce (*Picea abies*) and pine (*Pinus sylvestris*). Isoprene emission from these species is much lower compared to that from *Salix* [11,30]. In the Southern part of Sweden, the common land cover is farmland. Commercial crops growing on agricultural areas in Sweden, such as wheat, also emit significantly lower rates of isoprene [11,23]. Hence, a land cover change from the traditional species to *Salix* plantations could alter the regional atmospheric chemistry leading to, e.g., increased levels of $O_3$. However, isoprene-emitting plants seem to tolerate $O_3$ better than other non-isoprene emitting plants, and in this sense, varieties such as Wilhelm may be more resistant if growing in areas with high prevailing $O_3$ concentrations [51–53].

The monoterpene ocimene was emitted by all varieties, but at different rates. For Tora, Wilhelm and Inger, ocimene contributed circa 25–69% of the total MT emissions, whilst it was a minor compound for Ester. Ocimene and linalool were the only MTs which showed light dependency in Tora, Wilhelm and Inger, but not in Ester, likely due to being emitted only in very low amounts. In Tora and Inger, ocimene emission did not show any clear indication of leveling out, even at the highest measured PAR values. Wilhelm, on the other hand, did not increase the emission of ocimene much after 1000 μmol $m^2$ $s^{-1}$, and linalool seemed to level out for Tora and Wilhelm when PAR was above 1000 μmol $m^2$ $s^{-1}$. To date, no study has reported a light dependent relationship for MT emissions from willow trees, because the focus of most studies has been on isoprene. Monoterpenes can be important for generating secondary organic aerosols [54–57]. Since Ester was the only variety that did not increase MT emissions with increasing light, this variety might be more suitable near urban regions with more solar irradiance to avoid impaired air quality.

Nerolidol was the most dominant SQT and, together with humulene, constituted 75% or more of the total SQT emissions. Ester and Inger emitted approximately the same amounts of SQTs. Both of these varieties are female hybrids suitable for warm climates, and Ester also originates from Inger, which probably explains the similarities. However, the fractions of the emitted BVOCs differed. For example, no camphene was emitted by Inger, while camphene contributed almost one third of the MT emissions of Ester. In addition, Inger was the only variety that had a clearly increased emission rate of caryophyllene when light availability increased.

Saplings emitted approximately 3–19 times more other VOCs than the trees that were 1-year-old. Younger plants are more vulnerable than mature ones, and one way to strengthen their survival could be to emit more BVOCs [58]. Tora, on plot 3, which had the same growing season as the saplings, emitted lower rates of other VOCs than the one year old Tora on plot 1, but higher than the saplings belonging to Tora on plot 2. The root system on plot 3 was established in 2003, which can be one reason why they differed in comparison to the saplings, since they had already a developed root system and trunk. The ratio between other VOCs and isoprene emission changed according to the aging of the trees. At the beginning of the season, the fraction of other VOCs exceeded the fraction of isoprene, but at the end of the season, the opposite was seen.

Compounds other than isoprene and MTs are rarely reported in studies on Salix trees, and only low emission rates have been observed for these compounds in the few studies that have [22]. However, the results of this study show that they should not be discarded, at least not for saplings. Hexanal, which was the most emitted other VOC, has been reported as an important compound in abiotic and biotic stress [59–61]. Irrespectively, the reason why leaf beetles attacked all of the other varieties but not Ester is unclear. No unique compound emitted by Ester was found. Benzaldehyde and xylenes have been reported as stress-induced compounds in trees [62]. Even though the emissions of compounds such as benzaldehyde, furfural, p-cymene, camphene and nerolidol were higher from Ester compared to from the other varieties, the major contributions to these emission rates were observed from the saplings belonging to Ester and not from the 1-year-old trees. Therefore, one suggestion why the insects avoid Ester could be that this variety has compounds or other substances stored within their leaves that are not emitted unless the surface layer is broken, making Ester less attractive for leaf beetles.

The average standardized isoprene emission for the whole season ($33.21 \pm 53.43$ µg $g_{dw}^{-1}$ $h^{-1}$) is in line with other studies that have measured emissions from Salix trees [11,23]. It is hard to make a straightforward comparison since the methods, soil, adaptation to local growing conditions, age and different clones are likely to affect the emissions, and all these pieces of information are seldom presented in studies. Wild growing Salix species will also probably have different emission rates compared to commercial managed species. According to Morrison et al. [23], standardized isoprene emission from Salix trees can be more than 100 µg $g_{dw}^{-1}$ $h^{-1}$ but many emission rates range from 20 to 50 µg $g_{dw}^{-1}$ $h^{-1}$.

The standardized average MT emission rate was $4.40 \pm 2.05$ µg $g_{dw}^{-1}$ $h^{-1}$. The time of the year has been shown to influence the emission rate, and other studies have reported that Salix trees are prone to emit higher concentrations of MTs when they recently have had their bud break [23,63]. The study done by Ghelardini et al. [64] showed that the day of bud burst for Salix can vary between seasons, and differ for different varieties [65]. For the trees studied in Ghelardini et al. [64], it took up to 260 degree days of T > 0 °C since the first of March to have a bud burst. This value was reached by the middle of April for plots 1 and 2, and by the end of April for plots 3 and 4, when counting degree days in the same way as in their study. The first campaign in this study was started by the end of May for plots 1 and 2, and in the middle of June for plots 3 and 4, which makes it unlikely that the observed emissions included any enhanced emissions of MTs close to the bud break. Besides, saplings planted on plots 2 and 3 had developed their leaves before they were put in the ground, and therefore, no elevated MT emissions were expected from them due to the changing processes during bud break and leaf development.

The sesquiterpenes were the group that contributed least to the total BVOC emissions. The standardized emissions were $2.51 \pm 2.03$ µg $g_{dw}^{-1}$ $h^{-1}$. Sesquiterpenes are, in general, less studied when measuring emissions from Salix. Toome et al. [66] observed emissions of α-copaene, (E,E)-α-farnesene and α-murolene from rust-infected leaves, but not from control leaves. Emissions of α-copaene and α-farnesene have also been seen for wild growing Salix species [67]. α-farnesene was emitted from all varieties in this study, but no visible sign of rust was seen from the measured leaves.

## 5. Conclusions

In this study, four different Salix varieties (Tora, Wilhelm, Ester and Inger) were studied in southern Sweden. The emissions of BVOC, net assimilation rates and water use efficiency were compared. The varieties were exposed to similar light levels in the leaf chambers to be able to focus on the variation between the varieties.

The measured isoprene emissions from Wilhelm were three times higher than those from Tora, a genetically related species, but this difference was not statistically significant. This outcome emphasizes the complexity behind BVOC emissions, and plants that are more closely related do not necessarily respond in the same way. To be able to fully understand emissions of BVOCs, factors as production pathways, the environment and stress factors need to be taken into account. These

parameters are preferably studied in laboratory experiments rather than out in the field. The results from this study show that Tora is a low emitter of isoprene, and it is suggested to be the best candidate near polluted areas, where the potential for, e.g., $O_3$ formation is higher. Tora, Wilhelm and Inger had equally good A, and are consequently all suitable for growing as SRC in southern Sweden or similar climatic environments. Ester, which had lower A but higher WUE than the others, might be more appropriate in warm and dry areas. A clear difference was observed for the non-terpenoid emissions when comparing tree ages. Saplings emitted rates several times higher than those from the one year old trees. Particularly, the average emissions of hexanal were high, but benzaldehyde and octanal also showed higher rates for some of the young varieties, which may strengthen the defense system for the more sensitive younger trees [58–60].

Even if the outcomes from this study are related to local environmental issues, they need to be considered from a broader perspective. These kinds of biofuel plantation exist in many places in Europe, which could affect the environment for many people if the plantations are close to polluted areas. In addition, since BVOCs also act as precursors to SOA and cloud formation, they will likely have a regional impact as well [54,68,69]. Finally, the results from this study point out that both variety and age should be considered in modelling when scaling up BVOC emissions to better estimate regional budgets.

**Supplementary Materials:** The following are available online at http://www.mdpi.com/2073-4433/11/4/356/s1, Figure S1: Experimental setup in the field. Figure S2: Chromatogram of a sample in GS-MS. Figure S3: Chromatogram of a background sample in GS-MS. Figure S4: Thermopluviograms for plot 1–4. Table S1: Information about the different plots. Table S2: Average T, PAR and GDDs for all campaigns. Table S3: Values used for the fitted curves in Figures 6–9. Table S5: Emissions of BVOCs for first growing season trees.

**Author Contributions:** Conceptualization, T.H. and T.K.; methodology, T.H.; software, T.H.; validation, T.K. and T.H.; formal analysis, T.K.; investigation, T.K.; data curation, T.K. T.H. and R.R.; resources, T.H. and R.R.; writing—original draft preparation, T.K.; writing—review and editing, T.K., R.R. and T.H.; supervision T.H.; project administration, T.K. and T.H.; funding acquisition, T.H. All authors have read and agreed to the published version of the manuscript.

**Funding:** The project was funded by FORMAS (2012-727).

**Acknowledgments:** We thank Sten Segerslätt and Stig Larsson at European Willow Breeding AB for supporting and the permission to do the measurements at the two plots in Billeberga. We also thank Anders Jonsson and Per-Olof Andersson in Grästorp for their support and to make it possible for doing the measurements here. Finally, we thank Ida Vedel-Petersen for performing the GC-MS analysis.

**Conflicts of Interest:** The authors declare no conflict of interest. The funders had no role in the design of the study; in the collection, analyses, or interpretation of data; in the writing of the manuscript, or in the decision to publish the results.

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
