# Peer review of "Variability of BVOC Emissions from Commercially Used Willow (Salix spp.) Varieties"

_atmosphere, doi:10.3390/atmos11040356_

Round 1
Reviewer 1 Report
Variability of BVOC Emissions from Commercially Used Willow (Salix Spp.) Varieties
by Tomas Karlsson, Riikka Rinnanand Thomas Holst
The manuscript presents in detail the original results concerning the Biogene Volatile Organic Compound (BVOC) by different varies of willow proposed as a source of renewable energy.
The manuscript is well organized, with a multitude of experimental data which made it too massive.
There are a multitude of details which could be omitted or at last presented either as Supplementary material or as Appendices.
I would suggest the author to reduce significant the length of the paper, because, at its present form, the manuscript seem rather a detailed project report than a scientific paper.
Regardless these remarks, I consider the results presented in the manuscript as original, rather of local interest, but which, in a restrained form, worth being published.
My final recommendation is to be published, but after in a restrained volume.
Author Response
"Please see the attachment."

Reviewer 2 Report
General comments
This study deals with BVOC emissions from willow species used for biofuel production. The paper studies the emission at different areas and analysed the emission rate by age and willow species. The study remains of interest not only because of the characterization of these species but also regarding the key role of BVOC emissions in the secondary processing in the atmosphere. Nevertheless, some improvements need to be addressed in the manuscript.
Detailed comments
My main comments are related with the quality control and assurance control of the analytical method used here. More details need to be provided about the calibration method used for all the compounds analysed, detection limits, uncertainties, linearity and repeatability for these compounds will be much appreciated.
Sorbent tubes could also have limited breakthrough volumes for isoprene and other VOCs. The breakthrough test results for the compounds under analysis need to be addressed here.
Chromatograms obtained for some samples could be added to the supplementary material as well as some photos of the sampling dispositive used during field campaigns.
The authors mentioned the contamination of some species such as toluene, they also should explain the presence of other commonly observed anthropogenic VOC species such as p-xylene and benzaldehyde.
The manuscript is quite long and sometimes difficult to read. I would suggest avoiding unnecessary details in the results description or double representation of the data (tables and figures).
It is difficult to evaluate the significance and scientific soundnees of those emissions have in a regional and global context. How are BVOC Salix emissions compared with other commercial and non-commercial species? How lower or higher are there in comparison with other natural emission from forest and other green areas?
An estimation of secondary processing impacts will increase the quality and interest of this study.
Detailed comments
Some types were observed, please review the manuscript.
Table 1: I would suggest putting the age of species in months or decimal years
Table 5: xylenes are not oxygenated VOC
The quality and aligns and line breaks of the tables need to be improved. In some cases, is complicated to find the data for each line.
Table 7: Black shadows are very noisy, and the main information is lost because of that.
Table 8: I would suggest moving this table to supplementary material
Round 2
Reviewer 1 Report
Excepting some small errors regarding References format, everything is now OK. From my point of view, the manuscript could be published.
Author Response
Thank you so much again for reviewing our manuscript and helping us to improve it!
In this new edition some small changes have been made.
-The reference list have been edited according to the journal's guidelines.
-In addition, table 6 on page 11 has also been formatted due to reviewer 2's suggestion to make it clearer.
On behalf of all co-authors,
Tomas Karlsson
Reviewer 2 Report
Thanks for the answers.
I still believe that the quality of table 6 need to be improved.
Author Response
Thank you so much again for reviewing our manuscript and helping us to improve it!
In this new edition some small changes have been made.
-The reference list have been edited according to the journal's guidelines.
-In addition, table 6 on page 11 has also been formatted due to your recommendation to make it clearer.
On behalf of all co-authors,
Tomas Karlsson